# Causal Discovery for Modular World Models

**Anson Lei**
Applied AI Lab
University of Oxford, UK
`anson@robots.ox.ac.uk`

**Bernhard Schölkopf**
MPI for Intelligent Systems
Tübingen, Germany
`bs@tue.mpg.de`

**Ingmar Posner**
Applied AI Lab
University of Oxford, UK
`ingmar@robots.ox.ac.uk`

## Abstract

Latent world models allow agents to reason about complex environments with high-dimensional observations. However, adapting to new environments and effectively leveraging previous knowledge remain significant challenges. We present *variational causal dynamics* (VCD), a structured world model that exploits the invariance of causal mechanisms across environments to achieve fast and modular adaptation. VCD identifies reusable components across different environments by combining causal discovery and variational inference to learn a latent representation and transition model jointly in an unsupervised manner. In evaluations on simulated environments with image observations, we show that VCD is able to successfully identify causal variables. Moreover, given a small number of observations in a previously unseen, intervened environment, VCD is able to identify the sparse changes in the dynamics and to adapt efficiently. In doing so, VCD significantly extends the capabilities of the current state-of-the-art in latent world models.

## 1 Introduction

The ability to adapt flexibly and efficiently to novel environments is one of the most distinctive and compelling features of the human mind. It has been suggested that humans do so by learning internal models which not only contain abstract representations of the world, but also encode generalisable, structural relationships within the environment [5]. This latter aspect, it is conjectured, is what allows humans to adapt efficiently and selectively. Recent efforts have been made to mimic this kind of representation in machine learning. *World models* [e.g. 12] aim to capture the dynamics of an environment with a predictive model. Advances in latent variable models have enabled the learning of world models in a compact latent space [12, 35, 14, 9, 40] from high-dimensional observations such as images. Whilst these models have enabled agents to act in complex environments via planning [e.g. 14, 31] or learning parametric policies [e.g. 13, 12], structurally adapting to changes in the environment remains a significant challenge. The consequence of this limitation is particularly pronounced when deploying agents to environments, where distribution shifts occur. As such, we argue that it is beneficial to build structural world models that afford modular and efficient adaptation, and that *causal* modeling offers a tantalising prospect to discover such structure from observations.

Causality plays a central role in understanding distribution changes, which can be modelled as causal interventions [30]. The Sparse Mechanism Shift hypothesis [30, 6] (SMS) states that naturally occurring shifts in the data distribution can be attributed to sparse and local changes in the causal generative process. This implies that many causal mechanisms remain *invariant* across domains [29, 27, 38]. In this light, learning a *causal* model of the environment enables agents to reason about distribution shifts and to exploit the invariance of learnt causal mechanisms across different environments. Hence,

we posit that world models with a causal structure can facilitate modular transfer of knowledge. To date, however, methods for causal discovery [32, 26, 28, 8, 19] require access to abstract causal variables. These are not typically available in the context of world model learning, where we wish to operate directly on high-dimensional observations.

In order to benefit from the structure of causal models and the ability to represent high-dimensional observations, we propose *Variational Causal Dynamics* (VCD), which combines causal discovery with variational inference. Specifically, we train a latent state-space model with a structural transition model using variational inference and sparsity regularisation from causal discovery. By jointly training a representation and a transition model, VCD learns a causally factorised world model that can modularly adapt to different environments. The key intuition behind our approach is that, since sparse causal structures can only be discovered on abstract causal variables, training the representation and the causal discovery module in an end-to-end manner acts as an inductive bias that encourages causally meaningful representations. By leveraging the learnt causal structure, VCD is able to identify the sparse mechanism changes in the environment and re-learn *only* the intervened mechanisms. This enables fast and modular adaptation to changes in dynamics.

## 2 Background

In a complex environment with high-dimensional observations, such as images, learning a compact latent state space dynamics model of the environment has been successful in capturing scene dynamics [14, 9]. Given a dataset of sequences $\{(o^{0:T}, a_i^{0:T})\}_{i=0}^N$, with observations $o^t$ and actions $a^t$ at discrete timesteps $t$, a generative model of the observations can be defined using latent states $z^{0:T}$ as

$$p(o^{0:T}, a^{0:T}) = \int \prod_{t=0}^T p_\theta(o^t|z^t)p(a^t|z^t)p_\theta(z^t|z^{t-1}, a^{t-1})dz^{0:T}, \tag{1}$$

where $p_\theta(o^t|z^t)$ and $p_\theta(z^t|z^{t-1}, a^{t-1})$ are the observation model and the transition model respectively. The variational evidence lower bound can be written as

$$\mathbb{ELBO}(\theta, \phi) = \sum_{t=0}^T \mathbb{E}_{q_\phi(z^t|o^t)}\big[log(p_\theta(o^t|z^t))\big] - \mathbb{E}_{q_\phi(z^{t-1}|o^{t-1})}\big[\mathbb{KL}[q_\phi(z^t|o^t)||p_\theta(z^t|z^{t-1}, a^{t-1})]\big],$$

$$\tag{2}$$

where $q_\phi(z^t|o^t)$ is a learnable approximate posterior of the observations. See app.B for the derivation. Previous works have explored various functional forms of the transition distribution [e.g. 14, 35, 40]. Despite their success, these models cannot reason about changes in the environment as they cannot structurally utilise prior knowledge learnt from different environments under distribution shift.

To this end, we argue that imposing a *causal* structure on the transition model equips the learning agent with the ability to adapt to changes in a modular and efficient manner. A causal graphical model (CGM) [28] is defined as a set of random variables $\{X_1, ..., X_d\}$, their joint distribution $P_X$, and a directed acyclic graph (DAG), $\mathcal{G} = (X, E)$, where each edge $(i, j) \in E$ implies that $X_i$ is a direct cause of $X_j$. The joint distribution admits a causal factorisation such that

$$p(x_1, ..., x_d) = \prod_{i=0}^d p(x_i|PA_i), \tag{3}$$

where $PA_i$ is the set of parents to the variable $X_i$ in the graph. Each of the conditional distributions can be considered as an independent *causal mechanism*. In contrast to standard graphical models, CGMs support the notion of *interventions*. An intervention on the variable $X_i$ is modelled as replacing the conditional distribution $p(x_i|PA_i)$ while leaving the other mechanisms unchanged. Given the set of intervention targets $I \subset X$, the interventional distribution can be written as

$$p'(x_1, ..., x_d) = \prod_{i \notin I} p(x_i|PA_i) \prod_{i \in I} p'(x_i|PA_i), \tag{4}$$

where $p'(\cdot|\cdot)$ is the new conditional distribution corresponding to the intervention. The SMS hypothesis [30] posits that distribution shifts tend to correspond to sparse changes in a causal model. Causal mechanisms thus tend to be *invariant* across environments [29, 27, 39]. In this light, we argue that a *causal* world model can leverage the invariance within distribution shifts for fast adaptation.

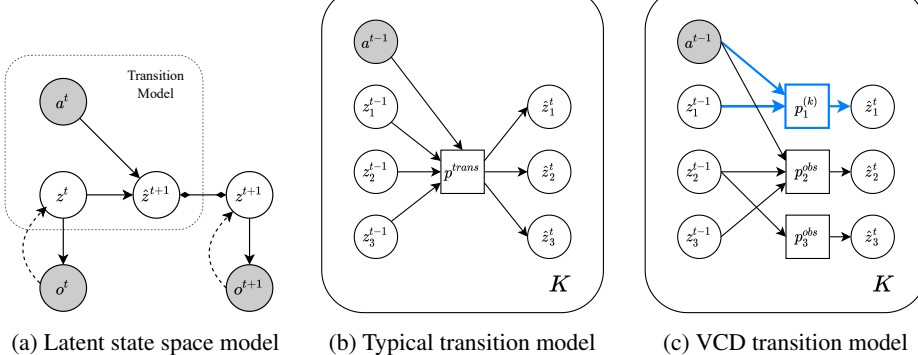

|(a) Latent state space model | (b) Typical transition model | (c) VCD transition model |

Figure 1: (a) The structure of latent state space models in one timestep. (b) The structure of latent transition models that are parameterised as fully connected neural network. (c) The causal transition model in VCD. Each variable has a separate conditional distribution $p_i$. Each mechanism is conditioned on a subset of the previous state. The blue lines highlight the environment-specific intervened mechanism, as opposed to the shared mechanisms corresponding to the black lines.

In order to learn a causal latent world model, we draw inspiration from causal discovery methods which learn causal structures from data. Specifically, We focus on method that are formulated as continuous optimisation [8, 19, 6] as these can be naturally incorporated into the variational inference framework. In particular, we follow the formulation in Differentiable Causal Discovery with Interventional data [8] (DCDI), which optimises a continuously parameterised probabilistic belief over graph structures and intervention targets. See Appendix C for further detail.

## 3    Variational Causal Dynamics

Similar to causal discovery with interventional data, variational causal dynamics (VCD) learns from action-observation sequences from an undisturbed environment, $(o_{(0)}^{0:T}, a_{(0)}^{o:T})$, and $K$ intervened environments, $\{(o_{(k)}^{0:T}, a_{(k)}^{0:T})\}_{k=1}^K$. The general approach of VCD follows the latent state-space model framework (Fig.1a), where a latent representation is jointly learnt with a transition model by maximising the ELBO. In contrast to existing approaches that parameterise the transition probability $p(z^t|z^{t-1}, a^{t-1})$ as a feedforward neural network (Fig.1b), VCD learns a *causal* transition model by utilising inductive biases from causal discovery. Importantly, our approach is grounded in the hypothesis that causal structures can only be discovered on semantically meaningful causal representations of the system. We therefore argue that training a representation and a transition model jointly to optimise a causal discovery objective can lead to a latent representation that affords causal transition models and is hence semantically meaningful.[1] Taking the view that shifts in the dynamics can be attributed to sparse causal interventions, we posit that a causal transition model facilitates modular adaptation to new environments by leveraging the invariance of causal mechanisms.

### 3.1    Causal transition model

In order to discover causal structure on transition dynamics, the transition model in VCD is designed to mimic the structure of a CGM (Eq. 4). The following is the transition model for environment $k$:

$$p^k(z^t|z^{t-1}, a^{t-1}) = \prod_i^d p_{(0)}^{(0)}(z_i^t|M_i^{\mathcal{G}} \odot [z^{t-1}, a^{t-1}])^{1-R_{ki}^{\mathcal{I}}} p_i^{(k)}(z_i^t|M_i^{\mathcal{G}} \odot [z^{t-1}, a^{t-1}])^{R_{ki}^{\mathcal{I}}}, \quad (5)$$

where $p^{(0)}$ is the transition model shared across all environments, $p^{(k)}$ is the interventional distribution specific to environment $k$, $M_i^{\mathcal{G}}$ is the binary causal masks that selects the causal parents for variable $i$, $R_{ki}^{\mathcal{I}}$ is a binary variable that is 1 when the variable $i$ is an intervention target in environment $k$. This formulation of the transition is designed around three main architectural features (Fig. 1c):

---

[1]The reader is referred to [22] for a discussion of the identifiability of causal representations in this setting.

- **independent mechanisms**. In contrast to fully connected models, the transition distribution for each latent dimension is parameterised by separate networks. This is motivated by the Independent Causal Mechanisms principle which posits that the causal generative process of the variables is composed of autonomous mechanisms [30]. In this work, each conditional distribution is parameterised as recurrent GRUs [10, 14].
- **sparse causal dependencies.** Following the structure of a CGM, we condition each variable only on its causal parents according to the learnable causal graph $\mathcal{G}$. This is modelled using the binary masks $M_i^{\mathcal{G}}$ which mask away non-parent nodes.
- **sparse interventions.** The intervention mask variables $R_{ik}^{\mathcal{I}}$ are introduced to model interventions. These variables act as switches between reusing a shared observational model, $p^{(0)}$, and an environment-specific interventional model, $p^{(k)}$. This is analogous to $p(x_i|PA_i)$ and $p'(x_i|PA_i)$ in Eq. 4.

## 3.2 Training

Similar to DCDI, the causal graph $\mathcal{G}$ and the intervention targets $\mathcal{I}$ are jointly trained with the model parameters. We parameterise the belief over the causal adjacency matrix $M^{\mathcal{G}}$ as a random binary matrix. Each entry $M_{ij}^{\mathcal{G}}$ follows a Bernoulli distribution with success probability $\sigma(\alpha_{ij})$, where $\alpha_{ij}$ is a scalar parameter and $\sigma(\cdot)$ is the sigmoid function. Similarly, a random binary matrix $R^{\mathcal{I}}$ is parameterised using the scalar variable $\beta_{ki}$ for each entry. Unlike DCDI, due to the existence of latent variables, we cannot directly maximise the data likelihood. Instead, we maximise the expected ELBO across all environments over causal graphs and intervention masks with sparsity regularisation,

$$L(\theta, \phi, \alpha, \beta) = \sum_{k \in [0,1,\ldots,K]} \mathbb{E}_{\mathcal{G},\mathcal{I}} \big[ \mathbb{ELBO}(o_{(k)}^{0:T}, a_{(k)}^{0:T}; \theta, \phi, \mathcal{G}, \mathcal{I}) - \lambda_G |\mathcal{G}| - \lambda_I |\mathcal{I}| \big], \quad (6)$$

where $\lambda$ are the hyperparameters for regularisation, the ELBO term is given by the expression in Equation (2), in which the transition model, $p_{\theta}^{(k)}(z^t|z^{t-1}, a^{t-1})$, is further factorised as in Equation (5). The gradients through the expectation terms are estimated using reparameterisation tricks [18, 20].[2] Crucially, by applying sparsity regularisation on the causal graph and the intervention targets, the model is encourage to learn representation which afford sparse dependencies and sparse changes across environments.

## 3.3 Adaptation

Due to the modular nature of the transition model, VCD can adapt to new, unseen environments by jointly inferring the intervention targets and the new model parameters for the intervened mechanisms. One way to implement modular transfer is to train VCD on trajectories in the new environment while fixing the trained parameters for the causal graph and the mechanisms in the undisturbed environment. The new parameters can be jointly trained in an analogous way to Eq. 6. Under the SMS hypothesis, only a small subset of the mechanisms need to be adapted in a new environment. As such, by leveraging past experience, VCD can adapt more quickly to environment change.

## 4 Experiments

In this section, we demonstrate that VCD is able to learn from multiple environments by learning a causal world model that explicitly captures the changes between the environments as interventions. We show that VCD is able to learn representation that reflects the ground-truth causal variables of the environments, and illustrate that, in an unseen environment, VCD can leverage past experience to perform modular adaptation, resulting in significantly improved data efficiency over the baselines.

**Setting.** We evaluate VCD on a simulated dataset of a 2-D multi-body system which contains four particles that affect each other via a spring or an electrostatic-like force. We consider changes such as strengthening or removing one of the springs, changing the mass of a particle, or constraining the position of a particle along the $x$ or the $y$ axis, which can be considered as interventions on one or more causal variables. The observation is given by rendering the environment as an image.

---

[2]For further implementation details, derivation of the lower bound, see Appendices B and C.

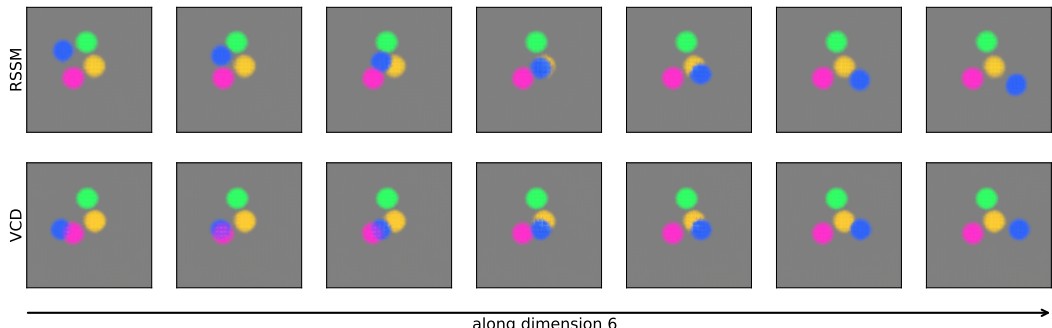

along dimension 6

Figure 2: Reconstructed images from samples along dimension 6 of the latent space. Compared to RSSM, VCD learns an axis-aligned representation where the blue particle moves horizontally.

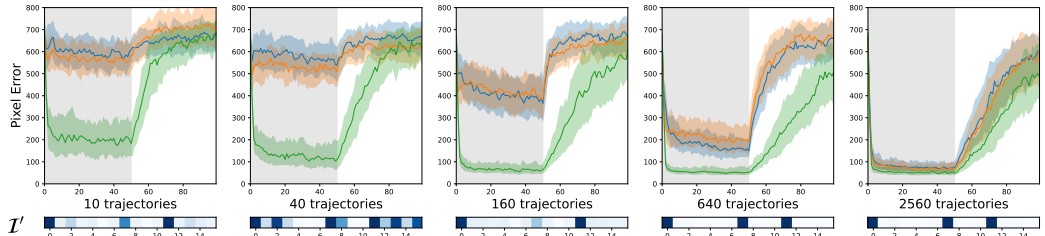

Figure 3: Rollout errors experiments where the models are trained on datasets of varying sizes (lower is better). The models receive observations for the first half of the trajectory (shaded), and perform latent space rollout for the rest. In the shaded areas, the one-step prediction error is reported. Non-pixel based error plots are also available in App. F. VCD significantly outperforms the other models with little data. The bar below each plot shows the learnt intervention targets in the new environment. VCD reuses most of the previously learnt mechanisms, as indicated by the sparsity in the learnt intervention targets.

We compare the performance of VCD against RSSM [14], a state-of-the-art latent world model. As RSSM does not support learning from multiple environments, we consider two adaptations of RSSM with different levels of knowledge transfer between environments: (1) **RSSM**, where one transition model is trained over all environments, i.e. maximum transfer of knowledge; and (2) **MultiRSSM**, where individual transition models are trained on each environment, i.e. no transfer of knowledge. We hypothesise that, compared to these two extremes of knowledge sharing, VCD is able to capture environment-specific behaviours whilst reusing invariant mechanisms via modular transfer.

**Representation.** One of the key hypotheses of this work is that jointly learning a representation and a transition model using causal discovery leads to causally meaningful representations. Here we examine the quality of the learnt latent space. Fig. 2 shows the image reconstructions of points drawn from a straight line along dimension 6 of the latent space. Since both models are initialised with the same encoder, this provides a qualitative intuition as to how the causal discovery inductive bias shapes the latent space. Compared to RSSM, VCD is able to learn an *axis-aligned* coordinate ($x$ coordinate) of the blue particle. Note that the motion of bouncing off the boundaries is only separable in the $x, y$ frame, implying that the dynamics in axis-aligned coordinates is sparser. A detailed analysis of the learnt representation and graphs is available in App. E.

**Adaptation.** We provide empirical evidence that VCD can adapt to a new environment with less data compared to the baselines by reusing learnt mechanisms in a modular fashion. All models are first pre-trained on 6 environments. We then collect datasets of different sizes in a previously unseen intervened environment where particle 1 is constrained horizontally. RSSM and MultiRSSM adapt to the new environment by optimizing the ELBO, with the difference that MultiRSSM instantiates a new transition model randomly and RSSM initialises the transition model using pre-trained parameters. VCD performs adaptation as described in section 3.3. Fig. 3 shows the rollout error of the models trained on datasets of varying size. Across all models, performance improves as the dataset grows. However, in contrast to RSSM and MultiRSSM, which overfits the dataset when the number of

trajectories is small, VCD is able to predict significantly more accurately. This is because VCD estimates the intervention targets and reuses trained modules that remain invariant.

## 5 Conclusion and discussion

In this paper, we propose VCD, a predictive world model with a causal structure that is able to consume high-dimensional observations. This is achieved by jointly training a representation and a causally structured transition model using a modified causal discovery objective. In doing so, VCD is able to identify causally meaningful representations of the observations and discover sparse relationships in the dynamics of the system. By leveraging the invariance of causal mechanisms, VCD is able to adapt to new environments efficiently by identifying relevant mechanism changes and updating in a modular way, resulting in significantly improved data efficiency.

## Acknowledgements

This research was supported by an EPSRC Programme Grant (EP/V000748/1). The authors would like to acknowledge the use of the University of Oxford Advanced Research Computing (ARC) facility in carrying out this work. http://dx.doi.org/10.5281/zenodo.22558. The authors thank Ceri Ngai, Jack Collins, Oiwi Parker Jones, Frederik Nolte and Jun Yamada for useful comments.

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

# A   Additional Related Works

VCD learns causal latent state-space models by combining variational inference and causal discovery. This section discusses related works in latent space world model learning, causal representation learning and causal discovery in relation to our work.

Predictive models of the environment can be used to derive exploration- [31] or reward-driven [12, 13, 14] behaviours. World models [12] train a representation encoder and a RNN-based transition model in a two-stage process. Other approaches [14, 40, 35] learn a generative model by jointly training the representation and the transition via variational inference. PlaNet [14] parameterises the transition model with RNNs. E2C [35, 4] and SOLAR [40] use locally-linear transition models, arguing that including constraints in the dynamics model yields structured latent spaces that are suitable for control problems. In a similar vein, the use of latent prediction models have also been explored in the context of video prediction [34, 11, 3]. Our proposed approach shares the general principle that latent representations can be shaped by structured transition mechanisms [1]. However, to the best of our knowledge, VCD is the first approach that implements a causal transition model given high-dimensional inputs in this context.

Causal discovery methods enable the learning of causal structure from data. Approaches can be categorised as constraint-based (e.g. [32]) and score-based (e.g. [15]). The reader is referred to [28] for a detailed review of causal discovery methods. Motivated by the fact that these methods require access to abstract causal variables, recent efforts have been made to reconcile machine learning, which has the ability to operate on low-level data, and causality [30]. Recent advances in this area include theoretical works on independent component analysis exploring the conditions under which disentanglement is possible [36, 17, 22]. Our current work situates within this broader context of causal representation learning, and aims to identify causally meaningful representations via the discovery of causal transition dynamics. [22] discusses the theoretical identifiability of a latent state-space model similar to ours. Whilst we do not make identifiability claims, our work builds on the inductive biases developed in [22]. To the best of our knowledge, VCD is the first model that focuses on the adaptation capabilities of causal models and empirically shows their applicability to image observations.

There has been significant interest in leveraging causal reasoning in world model learning [23, 24]. These approaches typically rely on predefined representations such as keypoints [24] or object slots [23], and attribute changes in the dynamics to unobserved confounders, which can be estimated from observations. Our approach differs in that we couple the learning of the representation and the model, which enables dynamics-aware discovery of causal representations. Similar approaches of jointly learning representation and model structure have also been explored in different contexts such as reinforcement learning [16] and temporally intervened sequences [25]. While the latter shares the general approach of learning sparse graphs in the latent space, it considers interventions as direct changes to the causal variables in each time step, whereas VCD formulates interventions as shifts in the dynamics *across environments*, which enables the capability of domain adaptation. AdaRL [16] considers few-shot adaptation under sparse mechanism shift (SMS) by attributing changes in the environments to low-dimensional environment-specific parameters. In contrast, by explicitly drawing from causal discovery, VCD models the dynamics of the scene as independent causal mechanisms. This, in turn, allows our approach to directly identify localised mechanism shifts between environments in an interpretable manner.

Another branch of causality-inspired work leverages the invariance of causal mechanisms by learning invariant predictors across environments [33, 29, 27, 39, 2]. This invariance has been studied in the context of state abstractions in MDPs [37], and invariant policies can be learnt via imitation learning from different environments [7]. In contrast, our approach models the full generative process of the data across different environments rather than learning discriminative predictors.

# B Derivations

Using the approximate posterior $q(z^{0:T}|o^{0:T}, a^{0:T}) = \prod_t q(z^t|o^t)$, the variational log lower bound for the latent state-space model (Eq. 2) can be derived from importance weighting and Jensen's inequality:

$$\log p(o^{0:T}, a^{0:T}) = \log \int \prod_{t=0}^{T} p(o^t|z^t)p(a^t|z^t)p(z^t|z^{t-1}, a^{t-1})dz^{0:T} \tag{7}$$

$$= \log \int \prod_{0}^{T} p(o^t|z^t)p(a^t|z^t)\frac{p(z^t|z^{t-1}, a^{t-1})}{q(z^t|o^t)}q(z^t|o^t)dz^{0:T} \tag{8}$$

$$\geq \mathbb{E}_{\prod q(z^t|o^t)}\left[\log\left(\prod_{o}^{T} p(o^t|z^t)p(a^t|z^t)\frac{p(z^t|z^{t-1}, a^{t-1})}{q(z^t|o^t)}\right)\right] \tag{9}$$

$$= \sum_{0}^{T} \mathbb{E}_{q(z^t|o^t)}\left[\log p(o^t|z^t) + \log p(a^t|z^t)\right]$$
$$+ \mathbb{E}_{q(z^{t-1}|o^{t-1})}\left[\mathbb{E}_{q(z^t|o^t)}\left[\log p(z^t|z^{t-1}, a^{t-1}) - \log p(z^t|o^t)\right]\right] \tag{10}$$

$$= \sum_{0}^{T} \mathbb{E}_{q(z^t|o^t)}\left[\log p(o^t|z^t) + \log p(a^t|z^t)\right]$$
$$- \mathbb{E}_{q(z^{t-1}|o^{t-1})}\left[\mathbb{KL}\left[q(z^t|o^t)||p(z^t|z^{t-1}, a^{t-1})\right]\right]. \tag{11}$$

Since the policy $p(a^t|z^t)$ is constant with respect to the model parameters, we omit this term and write the ELBO objective as

$$\mathbb{ELBO}(\theta, \phi) = \sum_{t=0}^{T} \mathbb{E}_{q_\phi(z^t|o^t)}\left[log(p_\theta(o^t|z^t))\right] - \mathbb{E}_{q_\phi(z^{t-1}|o^{t-1})}\left[\mathbb{KL}[q_\phi(z^t|o^t)||p_\theta(z^t|z^{t-1}, a^{t-1})]\right], \tag{12}$$

where $\theta$ is the model parameter and $\phi$ is the parameter for the approximate posterior. $\theta$ and $\phi$ are omitted henceforth to simplify notation. In VCD, the KL divergence term can be further decomposed by exploiting the structure of the transition model (Eq. 5), and the assumption that variables within each timestep are independent:

$$\mathbb{KL}\left[q(z^t|o^t)||p^{(k)}(z^t|z^{t-1}, a^{t-1})\right] \tag{13}$$

$$= -\int \log\left(\frac{p^{(k)}(z^t|z^{t-1}, a^{t-1})}{q(z^t|o^t)}\right)q(z^t|o^t)dz^t \tag{14}$$

$$= -\sum_{i=0}^{d}\int \log\left(\frac{p_i^{(0)}(z_i^t|M_i^{\mathcal{G}} \odot [z^{t-1}, a^{t-1}])^{1-R_{ki}^{\mathcal{I}}}p_i^{(k)}(z_i^t|M_i^{\mathcal{G}} \odot [z^{t-1}, a^{t-1}])^{R_{ki}^{\mathcal{I}}}}{q(z_i^t|o^t)}\right)q(z_i^t|o^t)dz_i^t \tag{15}$$

$$= -\sum_{0}^{d}\left((1-R_{ki}^{\mathcal{I}})\int \log\left(\frac{p_i^{(0)}(z_i^t|M_i^{\mathcal{G}} \odot [z^{t-1}, a^{t-1}]}{q(z_i^t|o^t)}\right)q(z_i^t|o^t)dz_i^t\right.$$
$$\left.+ R_{ki}^{\mathcal{I}}\int \log\left(\frac{p_i^{(k)}(z_i^t|M_i^{\mathcal{G}} \odot [z^{t-1}, a^{t-1}])}{q(z_i^t|o^t)}\right)q(z_i^t|o^t)dz_i^t\right) \tag{16}$$

$$= -\sum_{0}^{d}\left((1-R_{ki}^{\mathcal{I}})\mathbb{KL}\left[q(z_i^t|o^t)||p_i^{(0)}(z_i^t|M_i^{\mathcal{G}} \odot [z^{t-1}, a^{t-1}])\right]\right.$$
$$\left.+ R_{ki}^{\mathcal{I}}\mathbb{KL}\left[q(z_i^t|o^t)||p_i^{(k)}(z_i^t|M_i^{\mathcal{G}} \odot [z^{t-1}, a^{t-1}])\right]\right). \tag{17}$$

The KL terms can be computed analytically since the conditional distributions in the last expression are univariate Gaussian distributions. In training time, the gradients through the expectation terms in the ELBO is estimated by drawing a sample from the posterior distribution using the reparameterisation trick [20].

# C Implementation Detail

## C.1 DCDI and Graph Learning

This section covers the formulation of DCDI [8] and the graph learning method. These are subsequently used in the learning of VCD.

Given samples from an observed data distribution $P_X^{(0)}$ and $K$ intervened distributions $P_X^{(k)}$, DCDI optimises a probabilistic belief over causal graphs $\mathcal{G}$ and intervention targets $\mathcal{I}$. Specifically, these are encoded as random binary matrices, $M_\mathcal{G}$ and $R_\mathcal{I}$, where $M_{ij}^\mathcal{G} = 1$ implies that the edge $(i,j)$ is in the causal graph, and $R_{ki}^\mathcal{I} = 1$ implies that the variable $x_i$ is in the intervention targets in environment $k$. Each entry in $M^\mathcal{G}$ follows an independent Bernoulli distribution, parameterised by matrix $\alpha$ where $P(M_{ij}^\mathcal{G} = 1) = \sigma(\alpha_{ij})$. $R_{ki}^\mathcal{I}$ is similarly parameterised by $\beta$. Under this parameterisation, causal discovery can be formulated as maximising the expected data log likelihood with sparsity regularisation,

$$L(\theta, \alpha, \beta) = \mathbb{E}_{\alpha,\beta} \left[ \sum_{k=0}^{K} log[p_\theta^{(k)}(x_1^k, ..., x_d^k; \mathcal{G}, \mathcal{I})] - \lambda_G|\mathcal{G}| - \lambda_I|\mathcal{I}| \right], \tag{18}$$

where $p^{(k)}$ is the data likelihood under causal graph $\mathcal{G}$ and intervention targets $\mathcal{I}$ in the $k$th environment, as factorised in eq.(4). The conditional distributions are parameterised as feedforward neural networks with parameter $\theta$; $\lambda_{G,I}$ are hyperparameters to control sparsity. In the original DCDI framework, this is also subject to an acyclicity constraint. However, this is not neccessary in the context of our work as we assume there are no instantaneous causal effects (i.e., within a timestep).

The training objective for VCD can be viewed as a modified version of the DCDI objective, where the likelihood term is replaced with the ELBO (Eq. 12),

$$L^{VCD}(\theta, \phi, \alpha, \beta) = \mathbb{E}_{\alpha,\beta} \left[ \sum_{k=0}^{K} \sum_{t=0}^{T} \mathbb{ELBO}(\theta, \phi; \mathcal{G}, \mathcal{I}) - \lambda_G|\mathcal{G}| - \lambda_I|\mathcal{I}| \right], \tag{19}$$

Note that the expected number of edges in $\mathcal{G}$ and $\mathcal{I}$ given $\alpha$ and $\beta$ is simply the sum of the probability of each entry being one. Therefore, the training objective can be computed as:

$$L^{VCD}(\theta, \phi, \alpha, \beta) = \mathbb{E}_{\alpha,\beta} \left[ \sum_{k=0}^{K} \sum_{t=0}^{T} \mathbb{ELBO}(\theta, \phi; \mathcal{G}, \mathcal{I}) \right] - \lambda_G \sum_{ij} \sigma(\alpha_{ij}) - \lambda_I \sum_{ki} \sigma(\beta_{ki}). \tag{20}$$

The gradients through the outer expectation can be estimated using the Gumbel-Softmax trick [18]. To implement this, the ELBO term is evaluated with a sample of the causal graph using the following expression for each entry,

$$M_{ij}^\mathcal{G} = \mathbb{I}(\sigma(\alpha_{ij} + L_{ij}) > 0.5) + \sigma(\alpha_{ij} + L_{ij}) - stop\_gradient(\sigma(\alpha_{ij} + L_{ij})), \tag{21}$$

where $\mathbb{I}(\cdot)$ is the indicator function, $L_{ij}$ is a sample from the logistic distribution, and $stop\_gradient$ is a function that does not change the value of the argument but sets the gradient to zero. Samples for the intervention targets are similarly acquired. Note that the sample is used throughout each trajectory, i.e. the same sample graph and intervention targets are used for all of $T$ timesteps.

## C.2 Model Architecture

The encoders and decoders are parameterised as convolutional and deconvolutional networks from [12]. In the RSSM models, the transition models are parameterised as feedforward MLPs with two hidden layers of 300 hidden units. The recurrent module is a GRU with 300 hidden units. In VCD, to compensate for the fact each dimension in the latent space has a separate model, the number of hidden units in the GRU and MLP are reduced to 32 to avoid over-parameterisation. We found that initialising the encoders and decoders by pretraining them as a variational autoencoder helped with training stability for both RSSM and VCD.

The training objective is maximised using the ADAM optimiser [21] with learning rate $10^{-3}$ for mixed-state, and $10^{-4}$ for images. In both environments, we clip the log variance to $-3$, with a batch size of two trajectories from each of six environments with $T = 50$. In VCD, the hyperparameters $\lambda_\mathcal{G}, \lambda_\mathcal{I}$ are both set to 0.01. All models are trained on a single Nvidia Tesla V100 GPU.

# D Experiment Detail

**Interventions** The ground truth states of the multi-body dynamics environment is the $x$ and $y$ coordinates of each particle. The full list of possible interventions on the environment is provided in Table 1. Note that the forces between particle 1, 3 and 4 are proportional to their masses. Hence intervening on the mass of particle 1 and 3 also affect the dynamics of particle 4. In the experiments, all models are trained in the undisturbed environment and intervened environments 1, 5, 11, 14, 17.

In the mixed-state experiment, the observation function is a mixing matrix where each entry is drawn from a unit Gaussian distribution. In the image experiment, the observation is given by rendering the system to a $128 \times 128 \times 3$ image. In both experiments, the models are trained on a training set of 2000 trajectories from each of the six environments and evaluated on a validation set of 400 unseen trajectories.

Table 1: List of interventions

| ID | Intervention | Intervention targets |
|---|---|---|
| 1 | Remove spring between 1 and 2 | $x_1, y_1, x_2, y_2$ |
| 2 | Remove spring between 2 and 3 | $x_2, y_2, x_3, y_3$ |
| 3 | Increase mass 1 | $x_1, y_1, x_4, y_4$ |
| 4 | Increase mass 2 | $x_2, y_2$ |
| 5 | Increase mass 3 | $x_3, y_3, x_4, y_4$ |
| 6 | Decrease mass 1 | $x_1, y_1, x_4, y_4$ |
| 7 | Decrease mass 2 | $x_2, y_2$ |
| 8 | Decrease mass 3 | $x_3, y_3, x_4, y_4$ |
| 9 | Increase spring constant between 1 and 2 | $x_1, y_1, x_2, y_2$ |
| 10 | Increase spring constant between 2 and 3 | $x_2, y_2, x_3, y_3$ |
| 11 | Constrain movement of 1 to vertical only | $x_1$ |
| 12 | Constrain movement of 1 to horizontal only | $y_1$ |
| 13 | Constrain movement of 2 to vertical only | $x_2$ |
| 14 | Constrain movement of 2 to horizontal only | $y_2$ |
| 15 | Constrain movement of 3 to vertical only | $x_3$ |
| 16 | Constrain movement of 3 to horizontal only | $y_3$ |
| 17 | Constrain movement of 4 to vertical only | $x_4$ |
| 18 | Constrain movement of 4 to horizontal only | $y_4$ |

Table 2: The modified MCC scores for RSSM and VCD in both experiments

| Expriment | RSSM | VCD |
|---|---|---|
| Mixed-state | 0.728 | **0.975** |
| Image | 0.715 | **0.908** |

# E   Exploration of Learnt Latent Space and Causal Graph

### E.1   Representation quality

In this subsection, we explore the quality of the learnt latent space. The key hypothesis of our work is that training the representation jointly with the transition model to maximise a causal discovery objective serves as an inductive bias that helps to structure the latent space in a causally meaningful way. We provide the mean absolute correlation coefficient score for the representations for both RSSM and VCD.[3] We see that in both cases, training with the sparsity regularisation in VCD achieves a higher MCC score, which confirms that VCD learns a more disentangled representation.

The quality of the learnt representation is discussed in Section 4. Here we show reconstruction samples along all dimensions of the latent space in the RSSM and VCD representation. Note that since both encoders are initialised from the same pretrained VAE, the difference in the latent space arise because of the causal discovery

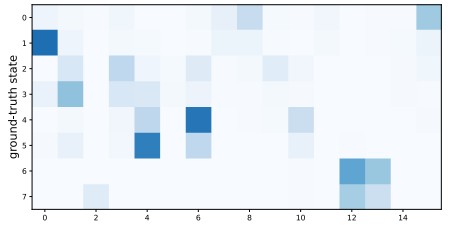 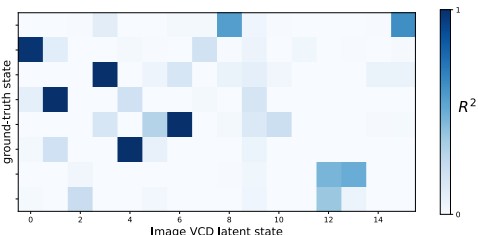

Figure 4: The $R^2$ of a linear regression between the ground-truth states and the learnt latent variables. Unlike the RSSM representation, VCD learns to focus on only one ground-truth state per dimension, as indicated by the fact that each column has only one main correlated state.

---

[3]Due to the fact that the ground-truth state dimension (8) is less than the latent state dimension (16), we modify the MCC score by calculating the mean of the top 8 scores out of the 16, effectively ignoring the latent variables that do not capture the state information.



Learnt Causal Graph            Learnt Intervention Targets



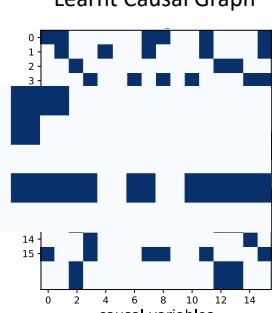
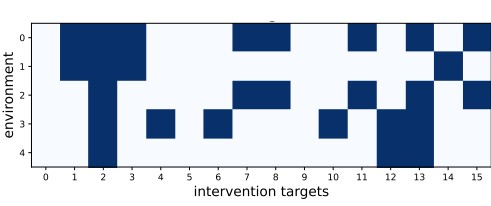

Figure 5: The learnt causal graphs and intervention targets for the experiment, obtained by binarising the learnt edge/ intervention target probabilities such that a blue square at $(i, j)$ implies $\sigma(\alpha_{ij}) > 0.5$, i.e. $i$ is a causal parent of $j$, or $j$ in intervened in environment $i$ respectively.

Table 3: Sparsity of learnt causal graphs and intervention targets. We compare the learnt graph with the ground truth causal graph by mapping each latent dimension to a ground-truth state as shown in Fig. 4. Note that the number of correct edges and the false positives do not sum to the number of edges in the graph because the edges corresponding to 'dummy' variables are ignored.

|  | # of edges | Correct Edges | Missed Edges | False Positives |
|---|---|---|---|---|
| causal graph | 73/288 | 22 | 4 | 13 |
| intervention targets | 31/80 | 10 | 0 | 17 |

objective in VCD. Fig. 7 shows the changes in the reconstruction when the RSSM representation is perturbed in each dimension of the latent space. Fig. 6 shows the same for VCD. As discussed in the main text, VCD learns a *axis-aligned* representation that affords a sparse causal graph. For example, dimension 1 and 3 captures the $y$ and $x$ coordinates of the green particle respectively. In contrast, RSSM learns a representation that is not axis-aligned. Fig. 4 plots the $R^2$ linear regression scores between the learnt latent variables and the ground-truth variables, showing that the VCD representation can in general disentangle individual states, as indicated by the more salient squares.

### E.2 Learnt causal graphs and intervention targets

In this subsection, we explore the quality of the learnt causal graph and intervention targets. Fig. 5 shows the learnt causal graph and intervention targets for the experiment. The sparsity of the learnt graph and targets is summarised in Table 3. VCD has identified 73 causal edges, out of 288 possible edges. Viewed in conjunction with the prediction performance results, this shows that VCD is able to learn a world model that is sparsely connected and affords modular parameter sharing *without* compromising on prediction accuracy.

VCD is able to identify a majority of the correct causal dependencies and *all* of the intervention targets. Upon further inspection of the learnt causal graph, we find that all of the seven missed edges correspond to the $1/||\delta \mathbf{x}||^2$ terms that scale the electrostatic-like forces. We hypothesise that the model cannot capture these dependencies as they are not as significant as the other forces.

## F  Non-pixel-based error

While pixel error is indicative of model performance in the short term, it provides limited clarity in long term predictions as two non-overlapping balls leads to the same pixel error regardless of their distance. In this section, we present an alternative latent space distance-based evaluation metric, Hit at 5 (H@5). This metric is defined by the proportion of episodes where the predicted latent representation is in the top-5 nearest neighbour set of the encoded ground-truth image. A H@5 score of 1 means all predicted latent states lie within a close distance to the ground-truth representation. Fig 8 shows the adaptation plot. These are consistent with the results shown in the main text.

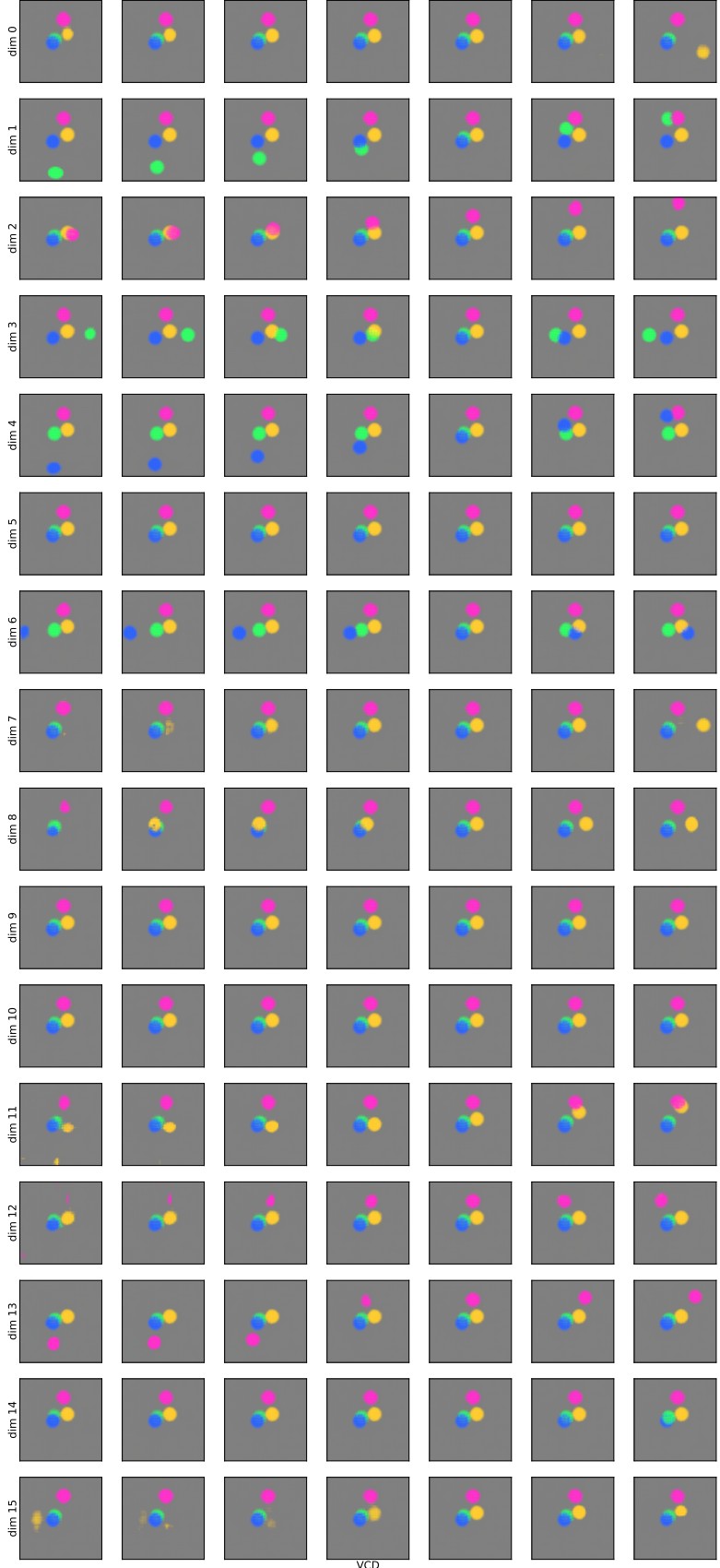

Figure 6: Visualisation of the learnt latent space in VCD.

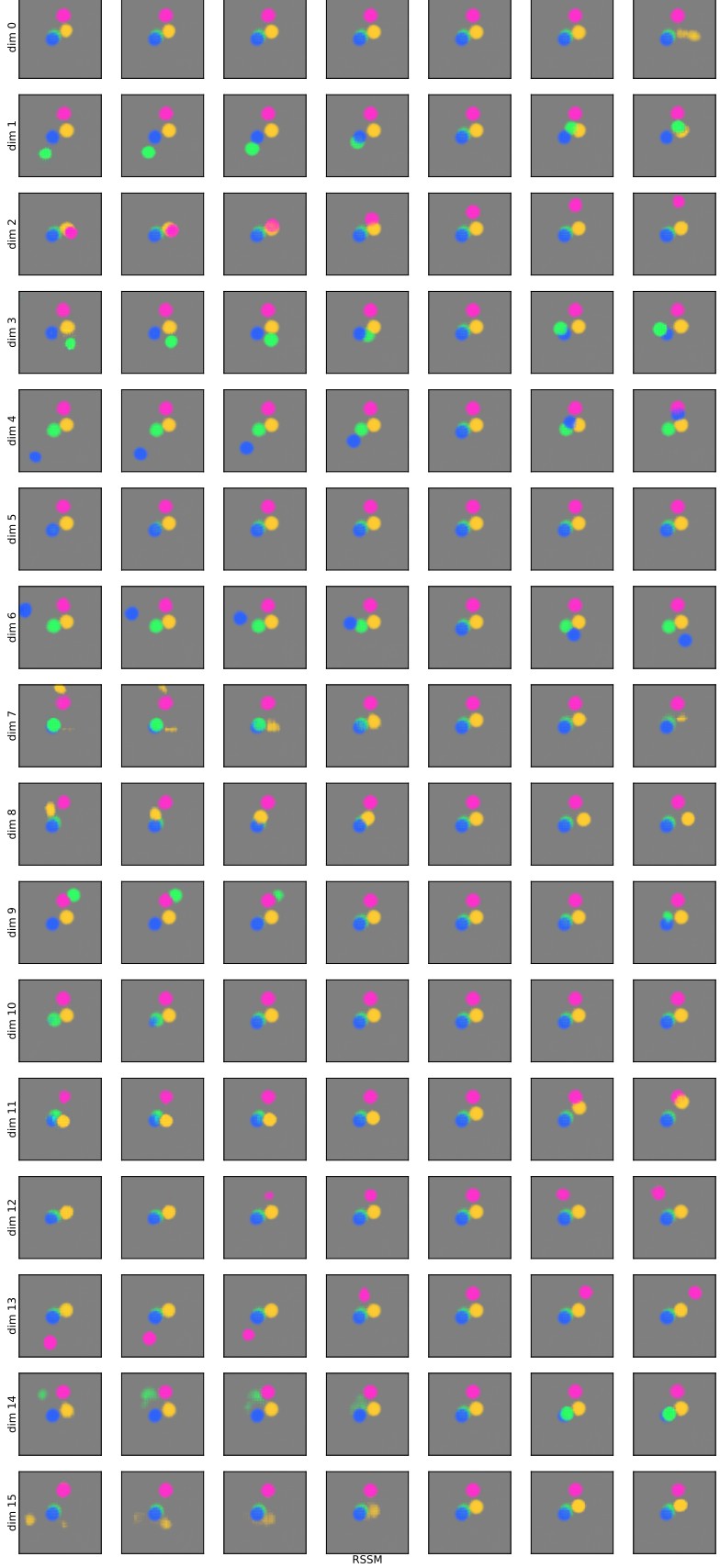

Figure 7: Visualisation of the learnt latent space in RSSM.

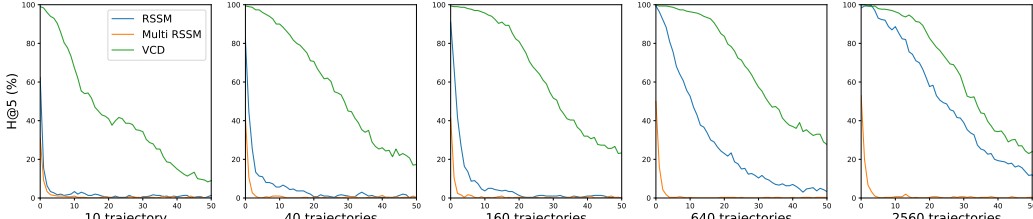

Figure 8: The adaptation plots with the H@5 scores.

