# OpenReview forum: "Causal Discovery for Modular World Models"
_NeurIPS.cc/2022/Workshop/nCSI — nCSI WS @ NeurIPS 2022 Poster_

### Official Review · Reviewer_tKt5 · 2022-10-12
**Interesting setup, but several presentation issues**

**Rating:** 1
**Confidence:** 2

**Review:**

The paper presents a method for learning causal mechanisms that can transfer across $k$ environments. The method learns a graph over time between latent variables, while also learning one observational and $k$ interventional distributions. By sharing the observational distribution, the model may learn variables that can easier generalize across environments. However, the paper has several issues in its presentation, which makes it hard to fully evaluate:
* The paper is not clear in separating its setting to previous work in causal representation learning, e.g. a longer discussion on the difference to [22,25] should be added. The goal is similar to these works, but the relation is in the current paper, e.g. the difference in the specific setup, is discarded.
* The paper several times states that it performs causal discovery and finds the causal structure. However, no proof has been given for the model actually learning the causal variables. Hence, the model may learn a completely different latent graph in which it finds a graph, but not the actually causal graph. This crucial distinction is dropped in the paper.
* The method seems to rely on the fact that the sparsity regularizer for $R^I$ is well tuned, since a too small value will allow the model to use the interventional distribution at all times. How is this value chosen? Was it finetuned for the specific experiment conducted in the paper? How stable is the model with respect to it?
* How does the method relate to different values of $K$, i.e. number of environments? What $K$ was used for the experiments?
* The dataset is very simplistic and the baselines of RSSM do not seem fair. A likely stronger baseline is simply using an iVAE where the prior is conditioned on the previous time step and the environment index, since both are available to the proposed model as well.
* Figure 2 seems very much cherry-picked and only gives a visualization of a single latent dimension. A quantitative result is missing. For example, in Figure 4, it gets apparent that some variables like the y-position of the pink ball was split across multiple dimensions (2, 13), and several other dimensions have issues (e.g. 12, 15).
* Figure 3: legends are missing. What are the green, orange and blue curves?
* Adaptation: using pixel error as in Figure 3 is not suitable metrics, since many things influence the reconstruction. Furthermore, it is unclear why the blue and orange (presumably RSSM) is so much higher from the start on when the observations are still given.

In conclusion, due to these major issues, I do not see this paper ready for presenting in this workshop. A major revision of the paper is needed to put it into perspective with previous work, clarify its claims with respect to causality, and improve the experimental setup.

Typos:
* Line 96: "In order to discover[-y-] causal structures on..."
* Line 106: space missing before citation.
* Line 110: "Sparse" capitalized, while "sparse" in 107 is lowercased.

---

### Official Review · Reviewer_raP8 · 2022-10-14
**Good idea, but is this actually causal?**

**Rating:** 2
**Confidence:** 2

**Review:**

The paper proposes the integration of causal discovery methods into the training of state transition models to build a causal world model. The method combines regular world models with DCDI and trains a latent variable model with interventional data (different environments)
Overall the paper is interesting and a good enough submission for a workshop.

Strengths:
- The application of causal modelling to world models is useful and yields good performance.
- The experiments seem to indicate that the method outperforms a relevant baseline.

Weaknesses:
- Given that the interventions and the state are unknown I am not sure whether this model can be correctly identified.
- It'd be good to clarify earlier that the interventions are unknown and what this means.
- It would be interesting whether one could also think of the interventions as an intervention on one or multiple separate global latent variable(s) that describes the behaviour of the system, e.g. the strength of the forces. Then the intervention becomes atomic and one could amortise the function over different environments. It'd be useful to discuss this.
- There are a few typos etc:
  - L97: "minic" instead of "mimic"
  - Eq. 6 - please explicitly explain what each lambda is used for
  - Fig 3: Please add a legend for the different colours. Also, I think it'd be useful to discuss the intervention matrix and how one could potentially evaluate that.

---

### Meta-Review · Area_Chair_H3ng · 2022-10-18

**Recommendation:** 2
**Confidence:** 4

**Metareview:**

The reviews are quite detailed and discuss the shortcomings of the paper. Mainly the presentation issues w.r.t positioning the paper in the context of causal representation learning is important.

Some of the issues raised by the reviewers will lead to a rejection in the conference however, as the goal of the workshop is to foster relationships and bridge research directions, I recommend acceptance. However, I strongly recommend that the authors please take all the comments and improve the paper for the final version.

---

### Decision · Program_Chairs · 2022-10-20

Accept (Poster)